# Leptin, Adiponectin, and Melatonin Modulate Colostrum Lymphocytes in Mothers with Obesity

**DOI:** 10.3390/ijms24032662

**Published:** 2023-01-31

**Authors:** Gabrielle do Amaral Virginio Pereira, Tassiane Cristina Morais, Eduardo Luzia França, Blanca Elena Guerrero Daboin, Italla Maria Pinheiro Bezerra, Rafael Souza Pessoa, Ocilma Barros de Quental, Adenilda Cristina Honório-França, Luiz Carlos de Abreu

**Affiliations:** 1Medical Clinic Department, Faculty of Medicine, Universidade de São Paulo, São Paulo 01246-903, Brazil; 2Postgraduate Program in Public Policies and Local Development, Escola Superior de Ciências da Santa Casa de Misericórdia de Vitória EMESCAM, Vitória 29045-402, Brazil; 3Departamento de Educação Integrada em Saúde, Universidade Federal do Espírito Santo, Vitória 29040-090, Brazil; 4Insittute of Biological and Health Science, Federal Univesity of Mato Grosso, Barra do Garças 78605-091, Brazil; 5Master of Public Health Program, School of Medicine, University of Limerick, V94 T9PX Limerick, Ireland; 6Laboratório de Delineamento em Estudos e Escrita Científica, Centro Universitário FMABC, Santo André 09060-870, Brazil

**Keywords:** adiponectin, breast milk, colostrum, hormones, leptin, lymphocytes, melatonin, obesity

## Abstract

Pregnancy complicated by obesity is associated with adverse triggered gestational and neonatal outcomes, with reductions in the subtypes of CD4+ T-lymphocytes representing the modulators of inflammation. It needs to be better established how maternal nutritional statuses impact the neuroendocrine–immune system’s action and affect the immunological mechanisms of the maternal–infant relationship via breastfeeding. This study examined the effects of maternal obesity on human colostrum lymphocytes and the intracellular mechanisms of lymphocyte modulation in the presence of leptin, adiponectin, and melatonin via cell proliferation; the release of intracellular calcium; and apoptosis induction. This cross-sectional study analyzed colostrum samples from 52 puerperal splits and divided them into overweight and eutrophic groups. Colostrum lymphocytes underwent immunophenotyping and cell proliferation by flow cytometry and intracellular calcium release and apoptosis assays by immunofluorescence in the presence or absence of hormones. Significant differences were considered when *p* < 0.05 by the chi-square or *t*-test. Maternal obesity reduced the population of T-lymphocytes and TCD4+ in human colostrum and proliferative activities (*p* < 0.05). These hormones restore lymphocyte proliferation to a level similar to the eutrophic group (*p* < 0.05). Leptin, adiponectin, melatonin hormones, and biological actions consolidated in the scientific literature also represent maternal and infant protection mechanisms via colostrum and the modulation of human colostrum lymphocytes.

## 1. Introduction

Obesity is highly prevalent in the global population. The aftermath of obesity during pregnancy is associated with neonatal complications that mainly arise from obesity-induced inflammation and changes in immune response mechanisms [1,2].

Obesity promotes changes in the profiles of cytokines and complement system proteins and immunoglobulins [3,4], hormones that regulate energy metabolism [5,6], and the functional activity of mononuclear colostrum cells by increasing intracellular calcium, promoting a higher basal rate of apoptosis [6,7]. In addition, it triggers changes in inflammatory mediators associated with adiposity [7,8].

In obese individuals, changes in the leukocyte population can be observed, emphasizing differences in the percentage of lymphocyte subtypes [9,10]. In addition, reductions in lymphocyte levels are also observed in the intrauterine environment in obese women, with reductions also being observed in CD4+ T lymphocytes, which are essential for modulating inflammatory responses and for maintaining a healthy pregnancy [1,11].

The repercussions of a pregnancy complicated by obesity are not restricted to the intrauterine environment. Maternal transfers during pregnancy and breastfeeding are important factors that modulate an infant’s immune system [12]. However, the effects of obesity during the pre-and postnatal period are still not fully understood. It is known that obesity promotes changes in the activity of human colostrum cells via mechanisms involving hormones such as leptin, adiponectin [7], and melatonin [6]; these hormones also promote the decreased release of intracellular calcium and decrease the apoptosis index in obese women [6,7].

Adiponectin and leptin are some of the adipokines in colostrum with concentrations that vary between eutrophic and obese mothers [5] and that differentially influence infant development during the first year of life [13]. These adipokines can also restore reduced macrophage activity in the colostrum from obese mothers; therefore, maintaining a balance between maternal adipokine levels can increase colostrum protection and contribute to reducing infection rates in infants breastfed by women whose pregnancy was affected by pre-gestational obesity [7].

Adiponectin is an appetite hormone that has important regulatory action on lipid and glucose metabolism. It stimulates food intake, and in infants, a high daily intake of this hormone can be achieved by intaking breast milk [5,13]. In addition, adiponectin has the potential to control inflammation in obese women due to its anti-inflammatory action [5,7]. Its concentrations increased in the colostrum of obese women, similarly to what has been observed for leptin [5].

Scientific evidence suggests an association between leptin in human milk and infant weight gain [14,15]. In addition to its relationship with energy metabolism, leptin represents a link between innate and adaptive immunity. It modulates cellular responses, including those of B and T lymphocytes. For example, leptin can increase the proliferation of naïve B and T lymphocytes and reduce the action of Treg cells, promoting a response with a pro-inflammatory Th1-like phenotype (which secretes IFNγ) instead of with anti-inflammatory Th2 (which secretes IL-4) and facilitates Th17 responses [16].

Although leptin exhibits pro-inflammatory action that has been described in the literature, it is noteworthy that biologically, there are compensatory mechanisms for the modulation of endogenous leptin levels. A recent study has shown that the hormone melatonin influences the modulation of endogenous leptin levels. MT1-type hormone melatonin receptors are important signaling pathways in leptin production, and when MT1 receptors are deficient, there is a 50% reduction in leptin receptor mRNA levels. However, these effects are reversible with the use of leptin, thus reducing weight and food intake [17].

Melatonin is predominantly produced in the pineal gland, and it controls circadian rhythms and affects energy homeostasis as well as the pathophysiological mechanisms that contribute to the development and maintenance of obesity and metabolic syndrome; in addition, it also influences the secretion of adipokines [18,19,20,21].

Scientific evidence emphasizes that in the presence of obesity, there is a reduction in the serum levels of melatonin [20,22,23]. However, melatonin is found in high concentrations in the colostrum of obese women [6]. Melatonin plays an essential role in children. It is present in colostrum and breast milk and follows circadian fluctuations, with peak concentrations found at night [24,25,26,27]. Melatonin ingested via breastfeeding plays an important role in synchronizing the biological rhythms of the mother–child dyad and increases the activity of phagocytes found in colostrum in their protection against infections [28,29,30]. In obese women, melatonin increases the protective activity of phagocytes and represents a possible maternal–infant protection mechanism against obesity [6].

Within this context, it can be observed that understanding the molecular mechanisms involved in obesity and in the relationship between maternal and child immunology represents a promising target for developing an intervention strategy against obesity and even against infections in neonates.

Changes in the immune system and maternal metabolism due pre-gestational obesity can likely affect the developing immune system of the neonate via changes in the lymphocyte population, promoting a greater risk of infection in the child. Nevertheless, due to the intense interaction between mother and child during breastfeeding, this risk may be minimized by the protective mechanisms that already exist in colostrum and in human milk, which involve the immunomodulatory actions of hormones such as leptin, adiponectin, and melatonin, which can regulate the activity of human milk lymphocytes. However, there are gaps in the scientific literature on the repercussions of maternal obesity on the functional activity of human colostrum lymphocytes.

Thus, this study analyzes the repercussions of maternal obesity on the proliferation of human colostrum lymphocytes and the intracellular mechanisms of lymphocyte modulation in the presence of adiponectin, leptin, and melatonin via intracellular calcium release and apoptosis induction.

## 2. Results

### 2.1. Characterization of the Sample

The characteristics of the analyzed population are shown in Table 1. It is important to note that the number of CD3CD4+ lymphocytes was higher in the eutrophic group (*p* < 0.05) (Table 1).

### 2.2. Molecular Analysis of Colostrum

The effects of obesity and its hormonal influence on colostrum lymphocytes were verified by the capacity for cell proliferation, functional activation via the intracellular release of calcium, and the possible effects on the induction of the apoptotic pathway for cell death.

The colostrum from mothers in the obese group showed lower cell proliferation (*p* < 0.05), but when cells were treated with the stimuli, cell proliferation increased. Nevertheless, in the stimuli presented here, proliferation decreased in colostrum cells from eutrophic mothers (Figure 1). The proliferative capacity of lymphocytes was only similar between the eutrophic and obesity groups in the presence of adiponectin (Figure 1a) and melatonin + leptin (Figure 1b) (*p* < 0.05).

Regardless of the maternal pre-gestational BMI, there were no differences in the spontaneous release of intracellular calcium by colostrum cells (*p* > 0.05), as described in Figure 2. The hormones leptin, adiponectin, and melatonin (Figure 2) reduced the intracellular calcium levels in cells with colostrum from the eutrophic group (*p* < 0.05). However, in the obesity group, in the presence of adiponectin and adiponectin + leptin, there was a decrease in the release of calcium from colostrum cells when compared to the same group treated with medium 199 (*p* < 0.05) (Figure 2a).

The rate of lymphocyte apoptosis did not change due to maternal excess weight (Figure 3). However, hormones leptin, adiponectin (Figure 3a), and melatonin (Figure 3b) reduced apoptosis in the lymphocytes from the eutrophic group after 2 h of incubation (*p* < 0.05).

The schematic representation summarizing the findings is illustrated in Figure 4.

## 3. Discussion

This study evaluated the interaction between adiponectin plus leptin and melatonin plus leptin based on the fact that in the literature, the actions of these hormones are consolidated regarding their role as mediators of inflammation. Adiponectin [31,32] and melatonin [33] have immunological actions with an anti-inflammatory profile, while leptin has more of a pro-inflammatory character [34,35].

Pre-gestational maternal obesity promotes changes in the immunological mechanisms of colostrum by reducing the percentage of T lymphocytes, especially CD4+ T, and also reduced the total proliferative activity of lymphocytes (Table 1). However, the hormones that regulate energy metabolism had the potential to restore cellular activity, mitigating the repercussions caused by maternal excess weight to recover cellular activity at levels similar to the group of eutrophic women. In the presence of the metabolic changes resulting from maternal excess weight, lymphocytes stimulated with adiponectin, leptin, and melatonin increased cell proliferation (Figure 1) without changing the levels of cell death by apoptosis (Figure 3). In contrast, adiponectin in the overweight group reduced intracellular calcium levels (Figure 2), thus highlighting its anti-inflammatory action in the overweight group.

It has been known for decades that human colostrum has a rich concentration of leukocytes ranging from 1 to 3 × 10^9^ cells/mL [36,37], and 5 to 10% of these cells are lymphocytes [37]. Among these, approximately 80% of the lymphocytes found in colostrum are T lymphocytes [38], and similar values were found in this study. Together, these cells make up an additional safety net for the infant that synergistically prevents pathogens from entering or that kills pathogens in order to not trigger uncontrolled inflammation [37,38,39,40].

Despite the established immunological benefits of breastfeeding, there are gaps in the in vivo mechanisms of maternal lymphocyte transfer to the infant. However, evidence from an experimental animal study indicates that the hematopoietically derived cells from breast milk survive in the infant’s gastrointestinal tract and remain until the infant is weaned. Furthermore, 80% of the cells transferred to an infant’s intestines are T lymphocytes and may be stored in regions called Peyer’s patches (P.P.s). Most of these cells (75%) are composed of TCD8+ lymphocytes, mainly because they exhibit high levels of the gut-homing molecules α4β7 and CCR9 but a reduced expression of the systemic homing marker CD62L. Thus, it is possible that there is an immunological compensation mechanism that protects the infant from the constant risks of oral infections during the postnatal period [39].

In this research study, it was observed that there were no changes in the levels of TCD8+ between the groups, but reductions were observed in the population of T CD4+ lymphocyte cells in the colostrum of obese donors (Table 1). In addition, a mixed culture of lymphocytes containing T and B lymphocytes presented a reduction in lymphocyte proliferation in the colostrum from the overweight group after 24 h of incubation with leptin, adiponectin, and melatonin (Figure 1). It should be emphasized that most lymphocytes present in colostrum are T lymphocytes [38] and that CD4+T lymphocytes are essential cells that have the potential to alter the markers of inflammation and are reduced in obesity [1,11].

There are gaps in the scientific literature on the repercussions of maternal obesity on human colostrum lymphocytes. However, evidence shows an increase in the number of TCD4+ lymphocytes that can be found in the intrauterine environment during pregnancy. Furthermore, it is known that there is a reduction in the proliferative response of lymphocytes relative to mitogenic stimulation in obese individuals [9,10]. This fact, possibly combined with changes in lymphocyte counts and the dysregulated expression of cytokines, is associated with obese children having a higher rate of infection than thin children [10]. Therefore, it is essential to modulate anti-inflammatory responses and to favor immune tolerance to prevent the maternal immune system from rejecting the fetus until’s delivery. In obese women, this cell population is reduced [1,11].

However, it is noteworthy that not all serum changes observed in obese women are transmitted to the colostrum [4,5]. In addition, changes in hormones such as adiponectin, leptin [7], and melatonin [6] in colostrum represent maternal–infant protection mechanisms that are capable of restoring the functional activity of cells to guarantee infant protection.

Our findings indicate that according to pre-gestational maternal BMI, adiponectin, leptin, and melatonin altered the proliferative activity of colostrum lymphocytes in different ways.

Importantly, lymphocytes express appetite-regulating hormone receptor profiles differently according to leukocyte subsets. This fact may partly explain the different actions of hormones on these cells [41]. Furthermore, metabolism-regulating hormones modulate T lymphocytes via interleukin-2 (IL-2) synthesis [42,43]. However, excess weight can cause a reduction in the number of lymphocytes expressing CD25. CD25 is a receptor, IL-2 binding site that is expressed on the surfaces of T lymphocytes [44]. This cytokine is a fundamental factor in the proliferation and activation of T lymphocytes [45]. Most lymphocytes in human milk are T cells [46]. Thus, the metabolic changes resulting from excess weight may cause different responses to adiponectin, leptin, and melatonin stimuli due to excess body weight via variations in the expression of IL-2 receptors.

There is evidence of metabolism-regulating hormones demonstrating proliferative activity. According to a study with breastfed animals who received adipokine supplementation, adiponectin showed increased lymphocyte proliferation, while leptin did not. Nonetheless, during breastfeeding, both adiponectin and leptin played a role in developing mucosal immunity early in life [47]. However, the proliferative activity of leptin under lymphocytes has also been reported in the literature; it can restore and increase the lymphocyte index despite obesity [48]. Melatonin conducts similar activities; indolamine also increases lymphocyte accumulation. In addition, it induces the proliferation of B lymphocytes [49] and T lymphocytes [50]. Melatonin acts directly on cell expansion, probably by binding to high-affinity receptors and IL-2 production, enhancing cellular immunity [51].

Although there are differences in lymphocyte proliferation as a function of maternal overweight, there were no differences in the lymphocyte viability found in the colostrum from eutrophic and overweight groups (Figure 1). Furthermore, according to maternal nutritional status, there were no significant differences in intracellular calcium release (Figure 2) and apoptosis (Figure 3). However, the stimuli showed anti-apoptotic effects, with a simultaneous reduction in intracellular calcium being observed in the eutrophic group.

The reduction in the endogenous levels of adiponectin is known to be associated with an increase in the apoptosis rate of the leukocytes found in peripheral blood [52]. Furthermore, the anti-apoptotic activity of this adipokine has been confirmed to control intracellular calcium levels, reduce the number of early apoptotic cells, and block the mitochondrial apoptosis process [53].

The anti-apoptotic effects of leptin that occurred via binding to the receptors upon the control of cellular apoptosis processes and intracellular calcium release are essential for controlling inflammatory responses, so the lymphocytes were treated with adiponectin, leptin, and melatonin. Intracellular calcium controls various cellular processes, including proliferation, differentiation, and cell death [54]. Moreover, the modulation of calcium dynamics and, consequently, of mitochondrial metabolism is fundamental for the survival of T lymphocytes under conditions of nutritional stress [55], suggesting that tested stimuli have no cytotoxic effects on colostrum lymphocytes; they have an action under the apoptosis process described in the scientific literature. Lymphocytes generate the positive regulation of Bcl-xL genes and the consequent inhibition of apoptosis [48]. Leptin treatment also attenuates apoptosis in lymphocytes [56]. However, in the presence of virulent pathogens, it induces apoptosis in this cell population, intensifying the immune response [57].

Melatonin also controls apoptosis in leukocytes, with the induction and elevation of intracellular calcium levels in the cytosol [58]. However, it also has anti-apoptotic effects on lymphocytes via mechanisms that reduce caspase-3 and 9 activities [59].

From the above, the differences between the action of the eutrophic and excess weight groups regarding the responses of lymphocytes can be seen. This difference suggests that the modulation of human colostrum lymphocytes is achieved by controlling the concentrations of the hormone’s adiponectin, leptin, and melatonin in the colostrum and by possible differences in the expression of receptors in these cells. Therefore, it would be relevant to analyze the levels of hormone receptors in these cells, representing a limitation of this study. In colostrum lymphocytes, among the stimuli tested, adiponectin was the only one that increased the proliferation index of lymphocytes in the overweight group that was dependent on reducing intracellular calcium levels. Leptin showed a better response when associated with adiponectin or melatonin, restoring the proliferative activity of lymphocytes to similar values to those found in the eutrophic group’s lymphocytes.

Thus, it is essential to highlight the need to maintain the balance between the hormones regulating the metabolism of the maternal organism in order to achieve adequate functional activity with respect to the mononuclear cells that make up human colostrum. However, we must emphasize that even with the differences in the responses of lymphocytes caused by maternal obesity, these cells provide immunological protection to the infant. Furthermore, the differences found in the immunomodulatory actions of adiponectin, leptin, and melatonin under the functional activity of lymphocytes may represent a defense mechanism present in the colostrum that ensures that the infant has an efficient immune system, as these cells come from an altered hormone environment and variable stimuli according to the presence of metabolic changes resulting from maternal obesity.

### Study Limitations

This study has some limitations; first, we should emphasize that the number of samples taken in each experiment is small because colostrum samples are difficult to collect, and the cells are not always present in sufficient amounts to complete all analyses when taken from the same woman; this limitation also restricts researchers from prioritizing the tests to be performed. Another limitation of the study is that the data were evaluated during the collection period and during only one milk maturation stage. Despite this, the tests indicate significant findings and should alert the scientific community to focus on the development of studies that emphasize other factors that may be involved when mothers with obesity are engaged in breast feeding.

## 4. Materials and Methods

### 4.1. Study Design and Participants

This was a cross-sectional study with laboratory analysis of human colostrum collected from volunteer postpartum women who gave birth in 2017 at the University Hospital of the University of São Paulo, Brazil. Both women who had a vaginal delivery or cesarean section were allowed to participate in the study, and the normal delivery rates in normal weight and obese women were 65.38% (n = 17) and 53.85% (n = 14), respectively. The study received approval from the Research Ethics Committee (CAAE 46643515.0.3001.0076).

The research study included clinically healthy postpartum women who were aged 18 to 35 years; had known or measured pre-gestational weight up to the end of the 13th gestational week; had a gestational age between 37 and 41^6/7^ weeks; did not have hepatitis, HIV, or syphilis; who performed prenatal care; and who did not have any dietary restrictions. In addition, we excluded postpartum women with gestational diabetes, twin pregnancy, and fetal malformations. Thus, the study population consisted of 52 women who were then divided into a eutrophic group (n = 26) and an obesity group (n = 26). Participants were distributed according to their pre-pregnancy body mass index (BMI): eutrophic (BMI between 18.50 to 24.90 Kg/m^2^) and obesity (BMI ≥ 30 Kg/m^2^) [60], as described in Figure 5. In addition, the anthropometric data of the puerperal women were collected from records for each woman that contained weights before and during pregnancy.

### 4.2. Colostrum and Cell Separation

Approximately 5 mL of colostrum was collected by hand milking during the day within 72 h after delivery. Subsequently, the samples were centrifuged for 10 min (160× *g*, 4 °C) for phase separation in the cell pellet, supernatant, and fat. Mononuclear cells were obtained using the Ficoll-Paque concentration gradient (Pharmacia, Uppsala, Sweden) and were resuspended in medium 199 (Gibco, Grand Island, NE, USA) [6,7]. Subsequently, the cells were incubated on a glass plate to determine phagocyte adhesion for 1 h in a CO_2_ oven at 37 °C. Non-adherent cells were removed, and the lymphocyte concentration was adjusted to 2 × 10^6^ cells/mL.

### 4.3. Immunophenotyping

Lymphocytes were labeled for 30 min with 5 µL of the respective monoclonal antibodies: anti-CD3 PerCP, anti-CD4 FITC, and anti-CD8 P.E. (B.D. Biosciences, San Jose, CA, USA). An isotype control (IgG1-FITC or IgG1-PE (B.D. Biosciences, San Jose, CA, USA) was used in all analyses, and cells underwent flow cytometry (FACSCalibur, BD Bioscience, San Jose, CA, USA).

### 4.4. Stimuli

Lymphocytes were incubated with medium 199 (control) or with the following human hormones: adiponectin (Sigma, St. Louis, MO, USA) at 100 ng/mL, leptin (Thermo Fisher, Carlsbad, CA, USA) at 100 ng/mL, melatonin (Sigma, St. Louis, MO, USA) at 100 ng/mL, adiponectin + leptin at 100 ng/mL, and melatonin + leptin at 100 ng/mL. The stimuli were endotoxin-free. The hormone concentrations were determined according to previous research with mononuclear cells from human colostrum taken from eutrophic and obese puerperal women [6,7].

### 4.5. Lymphocyte Proliferation

Lymphocytes were maintained in cultures in a CO_2_ incubator at 37 °C in the presence or absence of stimuli twice: at 2 and 24 h. Then, the cells were washed and stained with propidium iodide dye solution (440 µg/mL), Triton X-100 (5.5%), and EDTA (110 mM). A control without dye was used to determine the blank. Fluorescence intensity was determined by flow cytometry using FACSCalibur^®^ (B.D. Biosciences, San Jose, CA, USA), analyzing ten thousand events.

This method provides a proportional linear correlation between cell number and fluorescence intensity. The proliferation index was calculated by subtracting the fluorescence observed in cells after 24 h from the fluorescence intensity observed at the initial time [61].

### 4.6. Intracellular Calcium

Lymphocytes at a concentration of 2 × 10^6^ cells/mL were incubated with 50 µL of stimuli at 100 ng/mL and with 5µL of Fluo-3 AM solution at 1 mM (Sigma, St. Louis, MO, USA) for 2 h at 37 °C [6,7,62]. Cells were washed and resuspended in Hank’s Balanced Salt Solution (HBSS) with bovine serum albumin (BSA). Fluoroskan Ascent FL^®^ was used to measure fluorescence intensity with 485 nm excitation and 538 nm emission filters, and the results were described according to fluorescence intensity.

### 4.7. Apoptosis

After the lymphocytes had been incubated with the hormones for 2 h, apoptosis was analyzed using FITC Annexin V (B.D. Biosciences, Erembodegem, Belgium). Fluoroskan Ascent FL^®^ was used to measure fluorescence intensity using 485 nm excitation and 538 nm emission filters [6,7].

### 4.8. Statistics

Statistical analyses were performed with BioEstat^®^ version 5.0 software (Mamirauá Institute, Belém, Brazil). The Lilliefors normality test, *t*-test for independent data, and chi-square test were utilized; the results were presented as the mean (±standard deviation). Significant differences were considered when *p* < 0.05.

## 5. Conclusions

Pre-gestational maternal excess weight promotes a lower percentage of T CD4+ lymphocytes and increases the proliferative activity of lymphocytes in human colostrum. Proliferation was elevated in the overweight group after stimulation with adiponectin, leptin, and melatonin. These hormones showed different actions on lymphocytes according to maternal nutritional status. In the overweight group, adiponectin increased lymphocyte proliferation and then reduced intracellular calcium, maintaining basal apoptosis rates. The eutrophic group showed anti-inflammatory action, limited proliferative activity, decreased intracellular calcium levels, and demonstrated a lower apoptosis rate.

The data support the hypothesis that breastfeeding benefits the health of the mother and child, reducing body weight, controlling the inflammatory process, and reducing childhood infections. Thus, breastfeeding should be incentivized because it continues to be an effective public health strategy for combating the obesity epidemic and has real potential in providing protective effects against excess weight for both the mother and baby.

## Figures and Tables

**Figure 1 ijms-24-02662-f001:**
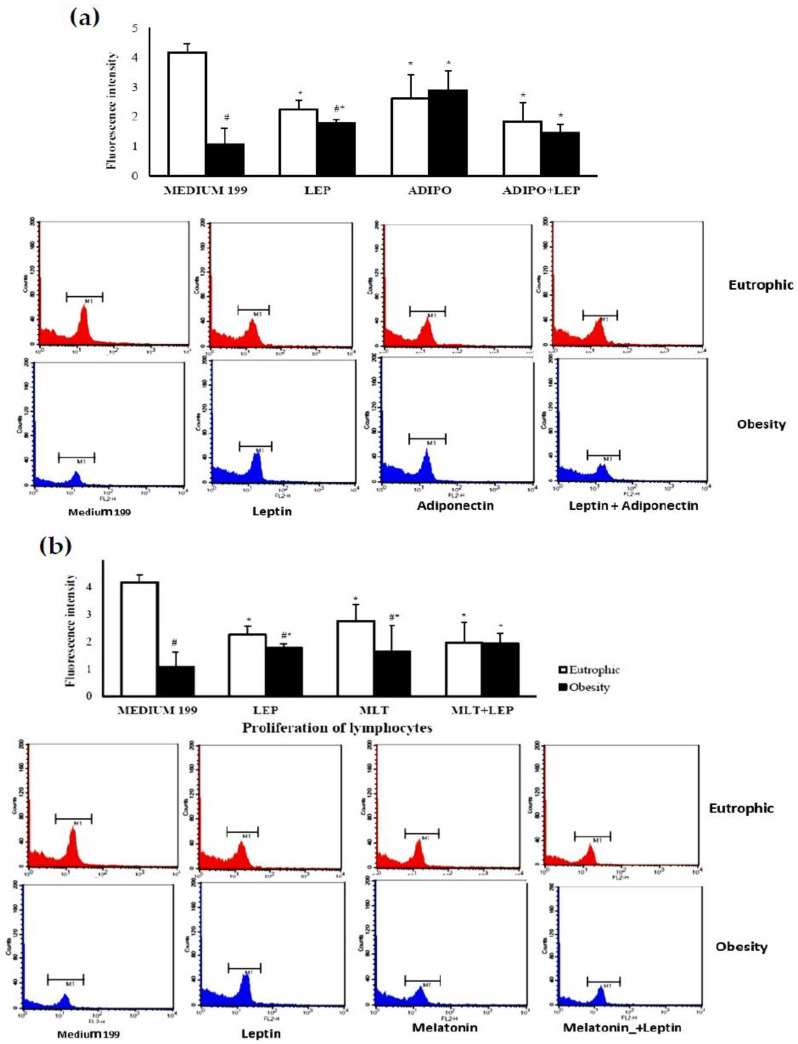
Fluorescence intensity of the proliferation of colostrum lymphocytes according to maternal nutritional status and stimulation by hormones leptin, adiponectin, and melatonin. After 24 h of incubation, colostrum lymphocytes proliferate in the presence or absence of the hormones: (**a**) leptin (LEP) and/or adiponectin (ADIPO) or (**b**) leptin and/or melatonin (MLT). The area represented by M1 represents the intensity of fluorescence of propidium iodide for each treatment. The results were evaluated by a *t*-test; data were expressed as Mean ± S.D. (n = 8 per group) and indicated as follows: * Statistical difference (*p* < 0.05) concerning the same group treated with medium 199. # Statistical difference (*p* < 0.05) intragroup (eutrophic x overweight considering the same treatment.

**Figure 2 ijms-24-02662-f002:**
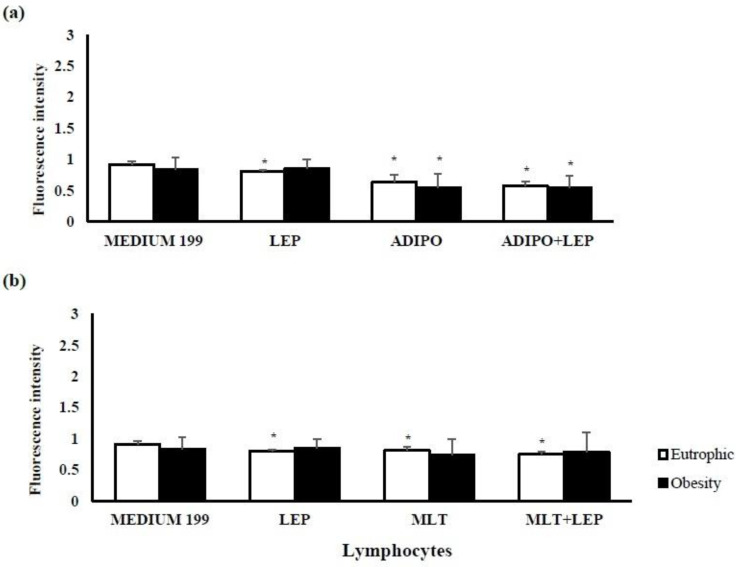
Detection of intracellular calcium released by lymphocytes from the colostrum of postpartum women according to maternal BMI and treatment with adiponectin, leptin, and melatonin hormones. Human colostrum lymphocytes release intracellular calcium in the presence or absence of (**a**) leptin and/or adiponectin or (**b**) leptin and/or melatonin. The results were evaluated by a *t*-test; data were expressed as Mean ± S.D. (n = 8 per group) and indicated as follows: * Statistical difference (*p* < 0.05) concerning the same group treated with medium 199 (control).

**Figure 3 ijms-24-02662-f003:**
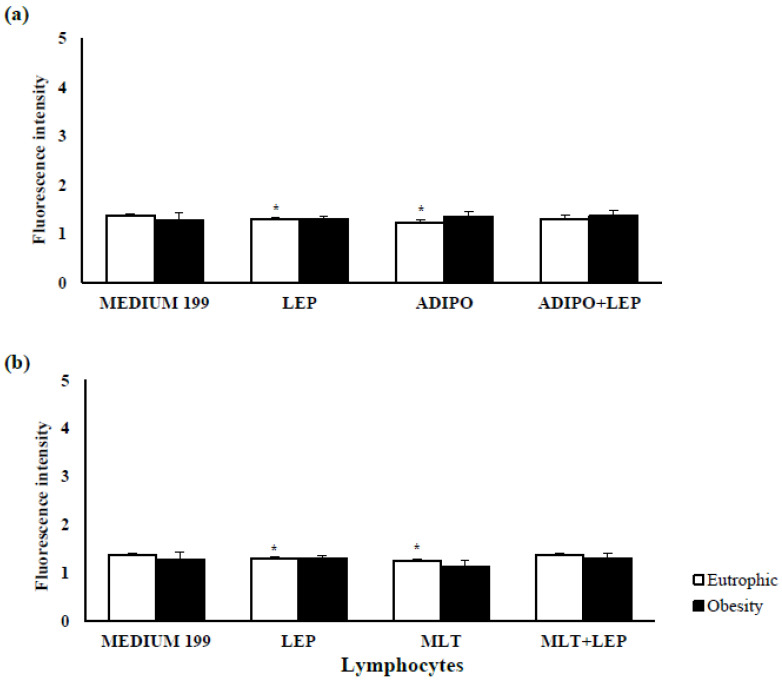
The action of the leptin, adiponectin, and melatonin hormone on the apoptosis index of human colostrum lymphocytes according to maternal nutritional status. Apoptosis index of human colostrum lymphocytes treated with hormones: (**a**) leptin and/or adiponectin or (**b**) leptin and/or melatonin. The results were evaluated by *t*-test; data were expressed as mean ± S.D. (n = 5 per treatment) and indicated as follows: * Statistical difference (*p* < 0.05) concerning the same group treated with medium 199 (control).

**Figure 4 ijms-24-02662-f004:**
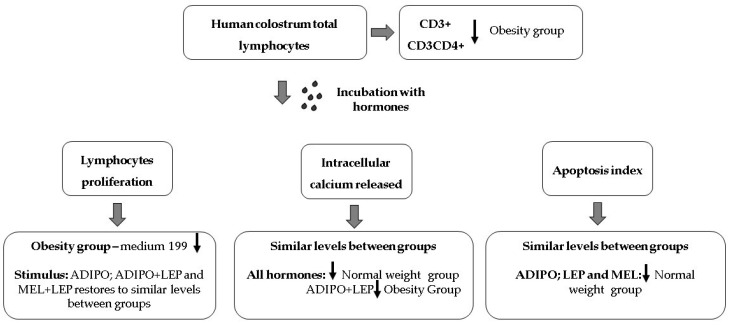
The schematic representation summarizes the main findings of immunophenotyping, lymphocyte proliferation, intracellular calcium release, and apoptosis. ADIPO = adiponectin; LEP = leptin; MEL = melatonin.

**Figure 5 ijms-24-02662-f005:**
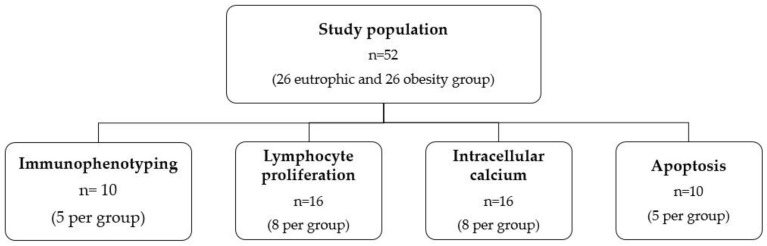
Flowchart of the number of colostrum samples used in each assay.

**Table 1 ijms-24-02662-t001:** Characterization of colostrum donors and their babies and the human colostrum T lymphocyte population.

Variable	Group		*p*-Value
	Eutrophic	Obese	
**Maternal** (n = 26 per group)			
Age (years) (Mean ± SD)	24.77 ± 5.54	26.17 ± 4.45	0.3327
Pre-Gestational Maternal Weight (Mean ± SD)	55.65 ± 6.82	86.87 ± 8.62 ^#^	<0.0001
Maternal height in meters (Mean ± SD)	1.59 ± 0.06	1.62 ± 0.07	0.1361
Final Gestational Weight (Kg) (Mean ± SD)	67.28 ± 8.12	94.22 ± 9.57^#^	<0.0001
Pre-gestational BMI in Kg/m^2^ (Mean ± SD)	21.84 ± 1.90	33.23 ± 2.35^#^	<0.0001
BMI at the end of pregnancy Kg/m^2^ (Mean ± SD)	26.59 ± 2.47	36.51 ± 2.92^#^	<0.0001
Gestational weight gain (Mean ± SD)	11.84 ± 4.52	8.20 ± 5.49^#^	0.0155
Gestational age in weeks (Mean ± SD)	38.65 ± 1.52	38.87 ± 1.45	0.6023
**Baby** (n = 26 per group)			
Baby’s sex—Female (%)	15 (57.69%)	16 (61.54%)	1.38 *
Birth weight in grams (Mean ± SD)	3143.85 ± 356.56	3316.20 ± 427.51	0.7852
Height in centimeters (Mean ± SD)	48.35 ± 1.84	48.59 ± 2.14	0.7041
**Lymphocytes** (n = 5 per group)			
CD3+ (Mean ± SD)	80.01 ± 13.72	39.72 ± 6.99 ^#^	0.0019
CD3CD4+ (Mean ± SD)	52.34 ± 10.73	22.35 ± 3.08 ^#^	0.0338
CD3CD8+ (Mean ± SD)	31.86 ± 12.33	17.37 ± 4.16	0.07

SD—Standard Deviation. # Statistical difference intragroup (*p* < 0.05) detected by *t*-test. * Chi-square test.

## Data Availability

The data supporting this study’s interpretations will be made available by authors if requested.

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
