# Peer review of "Leptin, Adiponectin, and Melatonin Modulate Colostrum Lymphocytes in Mothers with Obesity"

_ijms, 2023, doi:10.3390/ijms24032662_

Round 1

Reviewer 1 Report

Please find the attached file for comments and suggestions. In addition to the points mentioned in the document, the manuscript seems to be very discussion heavy. Please consider the additional experiments suggested in the document.

Reviewer 2 Report

The manuscript entitled “The leptin, adiponectin and melatonin modulate colostrum lymphocytes from mothers with obesity” by Virginio Pereira et al., addresses the hormonal modulation of lymphocytes in the colostrum in obesity. Thus, they isolate lymphocytes from colostrum from lean and obese women, and examine their population and their response to leptin, adiponectin, and melatonin and combinations of them in terms of proliferation, intracellular calcium, and apoptosis.

Despite the interesting and novel data provided in this study, which could contribute to our understanding of the maternal-infant relationship via breastfeeding, some experimental details should be better explained for correct interpretation of the results as well as some confusion exists in relation to the underlying molecular mechanisms.

Below follows specific questions and comments:

1)    Introduction section is unnecessarily extensive, and some paragraphs might be overlapping with the discussion part. On the other hand, results and material and methods should be described in more detail.

2)    Please describe the cohort shown in Table 1. Eutrophic women had a significantly higher gestational weight gain than obese women, how does that affect your findings? What kind of delivery had these women? What does it mean that CD3CD8+ lymphocytes were not affected by obesity?

3)    In Figure 1, the authors observed that lymphocytes from obese women proliferate more in response to the different treatments. However, the same treatments decreased proliferation in lymphocytes from eutrophic women, why?

4)    In Figure 3, how do you explain that lymphocytes from eutrophic women treated with a combination of hormones (adiponectin + leptin and melatonin + leptin) do not show significant results in apoptosis index, but it does when expose to each of these hormones separately?

5)    What is the reason why the combination treatment of adiponectin and melatonin or a combination of the three of them has not been tested?

6)    A schematic representation summarizing the findings would be required.

7)    What are the levels of these hormones in eutrophic vs. obese women during breastfeeding?

Reviewer 3 Report

Maternal overnutrition and maternal obesity are common phenomena in daily clinical practice and may pose severe long-term risks for the offspring. The authors propose to evaluate the leptin, adiponectin and melatonin in colostrum lymphocytes from mothers with obesity. They tested it in a very logical, straightforward way. However, from a technical point of view, the study has many drawbacks, especially regarding the composition of the text and the results.

1. The English grammar is awful and must be revised. 

2. It is unclear why the authors evaluated the release of intracellular calcium in the colostrum. The authors should explain it in the introduction and link it with the adipokines. 

3. In the Introduction, the authors did not explain the importance of evaluating adiponectin. It should be included a paragraph about it.

4. The Introduction is missing references, e.g. lines 101 to 108.

5. Why in table 1 are there not the p-values for "baby characterization" and for "CD3CD8+"? It is missing many p-values, which must be in the table even though the results are higher than 0.05.

6. Why did the authors not include the images of flow cytometry?

7. Do the authors evaluate the data distribution? That is very important to perform the correct test. The t-test is invalid for small samples from non-normal distributions. In the absence of normal distribution, the non-parametric Wilcoxon-Mann-Whitney (or rank sum) test is suggested as an alternative to the t-test, which does not rely on distributional assumptions.

Round 2

Reviewer 1 Report

The authors seem to have addressed the previous concerns and comments. Here are some additional comments and suggestions:

1. For Figure 1, could the authors please clarify the reason to include histogram and also what looks like a FACS plot? There seems to be "M1" written on all of these plots. Does it refer to M1 macrophages? Have the authors performed FACS analysis to show an overall shift in the phenotype of the macrophages?

2. For Figure 2, are the authors comparing the results of adipo group (both eutrophic and obese) with the medium 199 or they are comparing the eutrophic vs obese groups with respect to calcium release? The statistical difference sign indicates difference between groups of M199 and the respective condition, and the sign is in fact over the eutrophic group. However, the authors mention that the calcium release is lower only in obese group. According to the figure, for both eutrophic as well as obese group, the calcium release seems to be lower compared to M199. Could the authors please elaborate this?

3. In the description referring to Figure 2 (lines 157-162), please explain what is the significance of these findings briefly in results, and also include it in detail in discussion. The decrease in calcium release is also observed in adipo+lep group and not just adipo group as the authors mention here.

4. In line no. 195, the authors may need to also refer to Fig. 3b in addition to Fig. 3a. The overall effects on apoptosis seem to be very low. Is this expected or can the authors provide explanation for this?

Author Response

Dear Reviewer 1:

Thanks for your kind words about our research and the valuable feedback concerning our manuscript. We have carefully read and analyzed your remarks and answered every comment; see our answers in attached.

Sincerely

Prof. Adenilda C. Honorio-França

Reviewer 2 Report

The manuscript has been substantially revised in relation to the first version, strengthening its quality. The additional information included in the new version amend the issues raised and constitute a clear improvement that guarantee publication.

Author Response

Answers to reviewer 2

Dear Reviewer 2:

Thanks for your kind words about our research and the valuable feedback concerning our manuscript. Now the manuscript was revised by editing services MDPI.

Sincerely,

Prof. Adenilda C. Honorio-França

Round 3

Reviewer 1 Report

The authors have addressed previously mentioned concerns, as far as this reviewer is concerned.